# Cold-Hardy Grape Cultivar Winter Injury and Trunk Re-Establishment Following Severe Weather Events in North Dakota

**Andrej Svyantek [1], Bülent Köse [2], John Stenger [3], Collin Auwarter [1] and Harlene Hatterman-Valenti [1,*]** 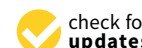

[1] Department of Plant Sciences, North Dakota State University, NDSU Dept. 7670, P.O. Box 6050, Fargo, ND 58108, USA; andrej.svyantek@ndus.edu (A.S.); collin.auwarter@ndsu.edu (C.A.)
[2] Department of Horticulture, Ondokuz Mayıs University, 55280 Samsun, Turkey; bulentk@omu.edu.tr
[3] Department of Agricultural and Biosystems Engineering, North Dakota State University, NDSU Dept. 7620, P.O. Box 6050, Fargo, ND 58108, USA; john.stenger@ndus.edu
[*] Correspondence: h.hatterman.valenti@ndsu.edu

**Abstract:** Extreme winter temperatures during the 2018–2019 dormant season contributed to trunk collapse and complete trunk death of numerous genotypes throughout a diverse grapevine planting in eastern North Dakota, USA. Through the early portion of the dormant season, 12 genotypes were screened to identify lethal temperature exotherms of primary buds; from these results, none were anticipated to be fully prepared to survive the −37 °C minimum temperature recorded in the region. Trunk collapse, death, and survival were monitored for 35 replicated genotypes. New trunks were retrained from suckers and monitored for growth following trunk removal. Only five genotypes exceeded 50% trunk survival at the end of the 2019 growing season, 'Valiant', 'King of the North', 'John Viola', 'Baltica', and 'Bluebell'. Following re-establishment, 'La Crescent' was the most vigorous genotype with the largest sucker circumference, sucker length, and internode length. Nearly all genotypes evaluated produced suckers with lengths approaching the high-wire trellis height (1.8 m), designating their potential for cordon retraining in 2020. Cumulatively, however, the lethal temperature exotherm results and the trunk survival examination indicate a harrowing need for investigation of new management practices (such as protected training systems) and the generation of new cold-hardy genotypes to enhance productivity under standard unprotected systems.

**Keywords:** cold-hardiness; retraining; differential thermal analysis; interspecific hybrid grapevine; *Vitis riparia*

## 1. Introduction

Low winter temperatures are one of the most critical environmental factors limiting productivity of grapevines in the Eastern United States [1–3]. Viticulture production is highly influenced by cold injury stemming from extremely low temperatures and drastic temperature fluctuations. The amount of grapevine cold damage is dependent on genotype, environment, and cultural practices [4,5]. Critical freezing events associated with acute climatic swings, such as polar vortexes, can contribute to nearly ubiquitous bud injury in *Vitis vinifera* genotypes [6]. While *V. vinifera* varieties are highly susceptible to temperatures below −20 °C, *V. riparia*-based interspecific hybrid grapevines are reported as resistant to temperatures below −35 °C [4,6]. Extreme winter injury leads to cane and trunk damage, incurring large and sustained economic losses for grape growers [6,7]. Even cold-hardy interspecific hybrid grapevines with diverse genomic contributions from *V. riparia* may be susceptible to injury under the most dire of Eastern US continental winter events [8,9].

Following cold damage, the grower must retrain the vineyard while minimizing costs [5]. Dead trunks of own-rooted vines require renewal with new shoots, typically suckers, arising at or just below the soil level [10,11]. This is a gradual process, requiring time for the grapevines and vineyards to re-achieve mature production capacity. Growers lose yield and profits during this recovery period. However, grapevines can return to producing fruit via trunk renewal in as little as 2 years [5,11]. In regions with periodic winter trunk collapse, the vineyard may be established with multiple trunks to reduce the risk of crop loss when a single trunk fails [11,12].

In areas where temperatures decrease to −20 °C or below, grapevines may be protected by covering with soil, snow, straw, or geotextile fabrics to reduce grapevine loss [5,8,10]. Burying of the vines under soil to prevent winter damage is a common application in many locations across the world [5,11,13,14]. However, this process is very labor intensive. At this time, no commercial grapevine producers in North Dakota actively practice any method of trunk winter protection. North Dakota grapevine production consists of cold-hardy interspecific hybrid grapevines, typically with a focus on wine grapes [9]. The short growing season and intense winter conditions have led to regional adoption of cultivars with high mid-winter freezing tolerance, with a high rate of *V. riparia* in their backgrounds.

Grapevines require acclimation and winter survival of compound buds, xylem, and phloem in order to consistently yield. Grapevine acclimation is a complex process in which green, non-hardy tissues, transition to woody, cold-hardy tissues following protein, carbohydrate, membrane, and hormonal-driven changes within cells of specific tissues [4]. Cane material (xylem and phloem) and dormant buds undergo dormancy acclimation processes as the growing season approaches an end, these include bud ripening and periderm development [4,8]. In fall and early winter, tissues increase in relative cold-hardiness before proceeding through deacclimation processes as spring warming conditions arise [4].

North Dakota has harsh winters characterized by extreme mid-winter low temperatures, high winds, and low annual winter snow accumulation [15]. During the 2018–2019 winter season, extreme freeze events experienced from 29 January to 31 January, with daily minimum temperatures of −34, −36.8, and −32 °C, contributed to cold-damage of many grapevine genotypes planted as a variety trial at the North Dakota State University Horticulture Research Farm (HRF). Responding to the severe winter injury, this study was initiated to develop an understanding of grapevine trunk replacement potential and growth characteristics of different germplasm following catastrophic vineyard failure. Further, goals included the identification of genotypes with potential to perform well for local farmers or for use as parents in regional grapevine breeding efforts.

## 2. Materials and Methods

In 2004, a cold-hardy hybrid grapevine variety trial was established at the NDSU HRF, near Absaraka, ND, USA. From 2004 to 2009, nearly 40 own-rooted genotypes (both released cultivars and breeding germplasm) were planted for evaluation; however, only 30 were retained through the initiation of this experiment in 2018. Between 2012 and 2017, additional own-rooted genotypes were planted for characterization in rows adjacent to the earlier established (2004–2009) variety trial planting; of these newly planted genotypes, five recently released cultivars were evaluated in this experiment. The entire unirrigated experimental vineyard plot was planted on a 0–2% slope in Warsing, sandy loam soil. Vines were trained to multiple trunks with spur-pruned, bilateral cordons on a single high-wire at 1.8 m above the soil level. Rows 83 m long were established in a North to South orientation with a spacing of 2.4 m between vines in-row and 3.0 m between rows. A 0.5 m weed-free strip was maintained below the vine rows using tillage combined with pre-emerge (Flumioxazin, Chateau ®, Valent USA, San Ramon, CA, USA) and post-emerge (Glufosinate, Rely®, BASF, Florham Park, New Jersey, USA) herbicide applications. Vineyard mid-rows were planted with red fescue (*Festuca rubra*). All grapevines genotypes were planted into four-vine experimental units in a randomized complete block-design with four replicates clustered by a period of planting. A total of 16 vines were examined per replicated genotype; an additional nine unreplicated genotypes (two vines per plot, one plot) were also examined

for trunk survival and sucker retraining metrics. Evaluations of grapevine cold-hardiness were conducted for 12 cultivars through the 2018–2019 dormant season, and evaluations of grapevine trunk survival and sucker retraining were conducted for 35 individuals in the 2019 growing season.

All weather data were recorded at a weather station that is part of the North Dakota Agriculture Weather Network located near Prosper, ND, USA approximately 18 km from the experimental site [16]. Depicted in Figure 1 are the minimum, maximum and average air temperatures from Sept. 2018 to May 2019, as well as the normal temperatures (based on 1971 to 2000). The main dormant season, regionally defined as the 7-month period spanning between 01 October and 01 May, was characterized by minimum daily temperatures below freezing (0 °C) for 185 out of 212 days. During this time, temperatures were at least −20 °C for 53 days, at least −25 °C for 35 days, and at least −30 °C for 8 days. The absolute minimum temperature of −36.8 °C was experienced 30 January.

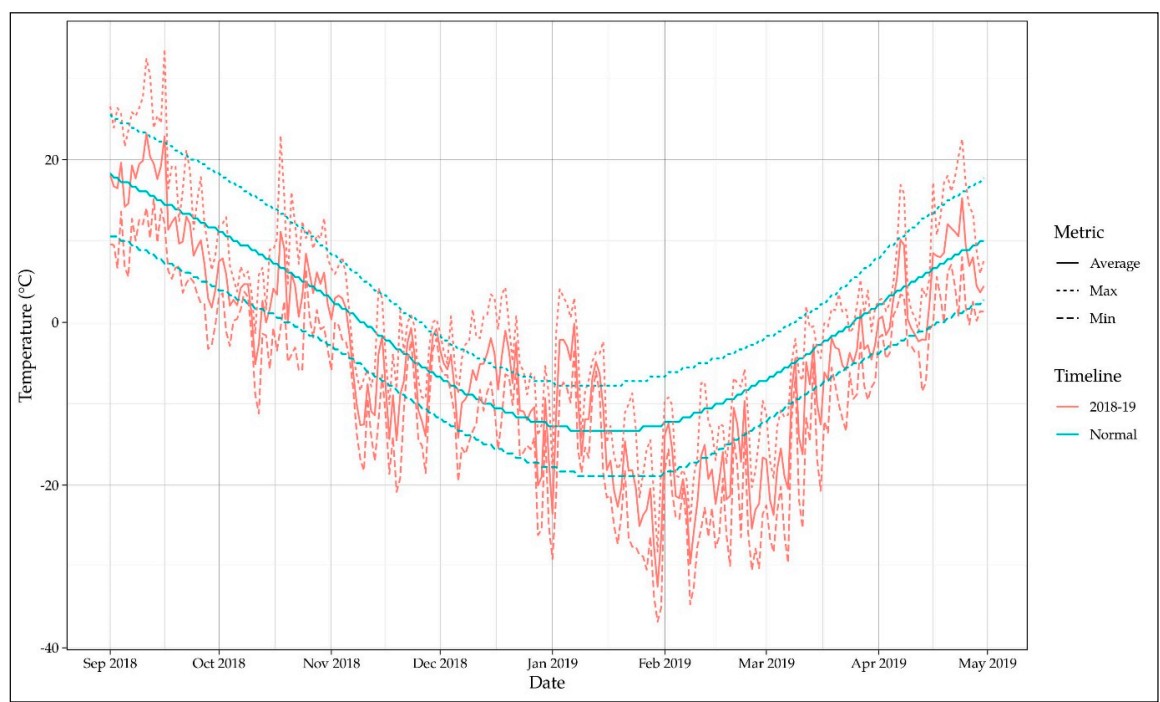

**Figure 1.** Normal (1971–2000) daily temperatures and temperatures from September 2018 to May 2019 recorded at the North Dakota Agriculture Weather Network weather station near Prosper, ND USA.

### 2.1. Differential Thermal Analysis

Dormant canes were collected twice monthly for a subset of 12 cultivars from early Nov. 2018 through mid Feb. 2019. Differential thermal analysis (DTA) was conducted on four field replicates per genotype. Buds sampled on each cane were collected from node positions three to eight, acropetally up from the base of the cane. Two-bud spurs were retained below the sampled cane to allow for production of fruit the following season.

Collection of lethal temperature exotherm responses (LTEs) were conducted similar to Mills et al. [17]. Canes were promptly cut into bud sections in the laboratory and placed on moist, pre-cut tissue squares (Kimwipes, Kimberly-Clark, Irving, TX, USA) within individual cells containing thermoelectric modules. LTE peak data were recorded though a Keithley Multimeter Data Acquisition System (model 2700, Tektronix, Inc., Beaverton, OR, USA). Freezing of buds was conducted within a Tenney Model T2C programmable freezer (Thermal Product Solutions, New Columbia, PA, USA). Following stabilization controlled by a Watlow Series 942 temperature regulator (Watlow Electric Manufacturing, St. Louis, MO, USA), the freezer was held at 4 °C for 1 h prior to experimental cooling with a reduction rate of −4 °C h$^{-1}$. Once the freezer reached a minimum temperature of −50 °C, the freezing cycle was

completed, and the programmable freezer was progressively warmed to 4 °C. The median LTE values (LTE$_{50}$) were identified manually with Bud Processor 1.8.0 software (Brock University, St. Catherines, ON, Canada).

## 2.2. Winter Injury

Following the 2018–2019 winter, damage to grapevines was first observed when dormant buds failed to break in spring for multiple genotypes. Grapevine sap flow from trunk cavitation and acute trunk splitting were observed sporadically (Figure 2). Subsequently, grapevines were monitored for survival of trunks on three separate occasions during the growing season: at budbreak (mid-May 2019), pre-veraison (mid-July 2019), and during dormancy (mid-November 2019). The total number of trunks per experimental unit were recorded and percentages of trunks that failed to break bud from any point (2019 shoot growth entirely absent from spurs, canes, cordons, and the length of the trunk to the soil level), collapsed mid-season (green shoots appeared early in season, but ceased growth), or survived through the entirety of the 2019 growing season were calculated. Due to regional renewal practices, at the start of the 2019 growing season, the number of established trunks present at each of the four sample grapevine plants within an experimental unit ranged from one to three.

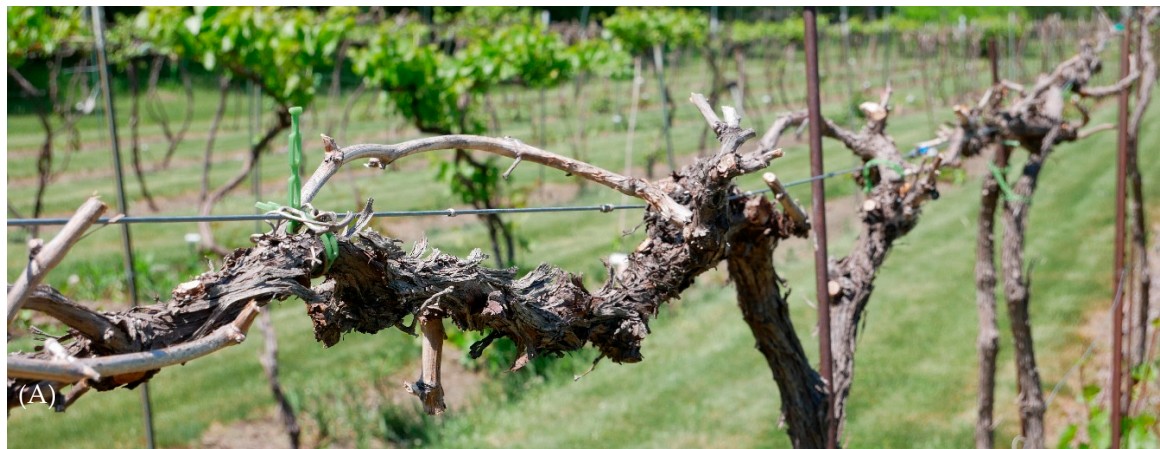

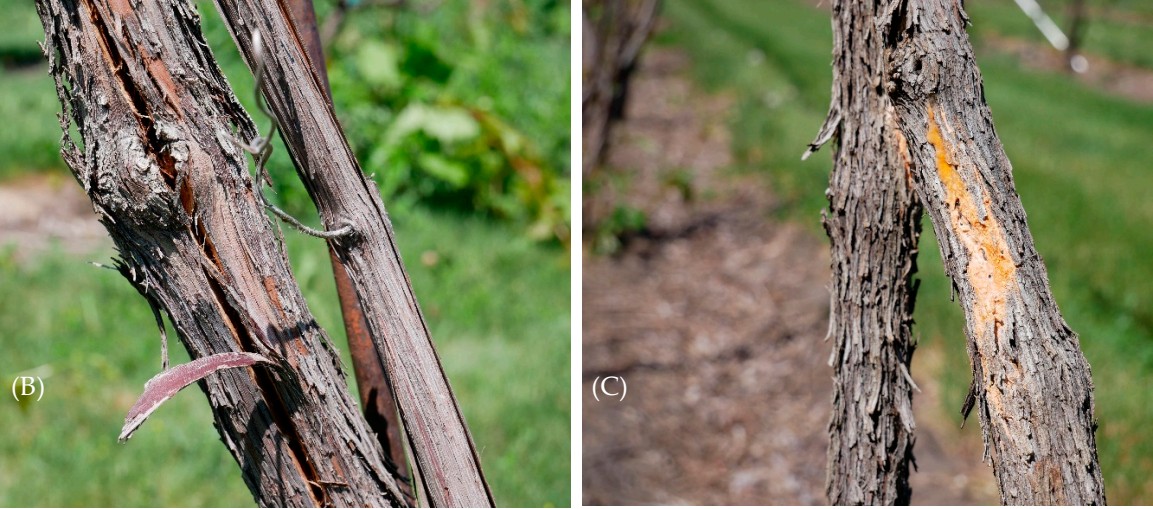

**Figure 2.** *Cont.*

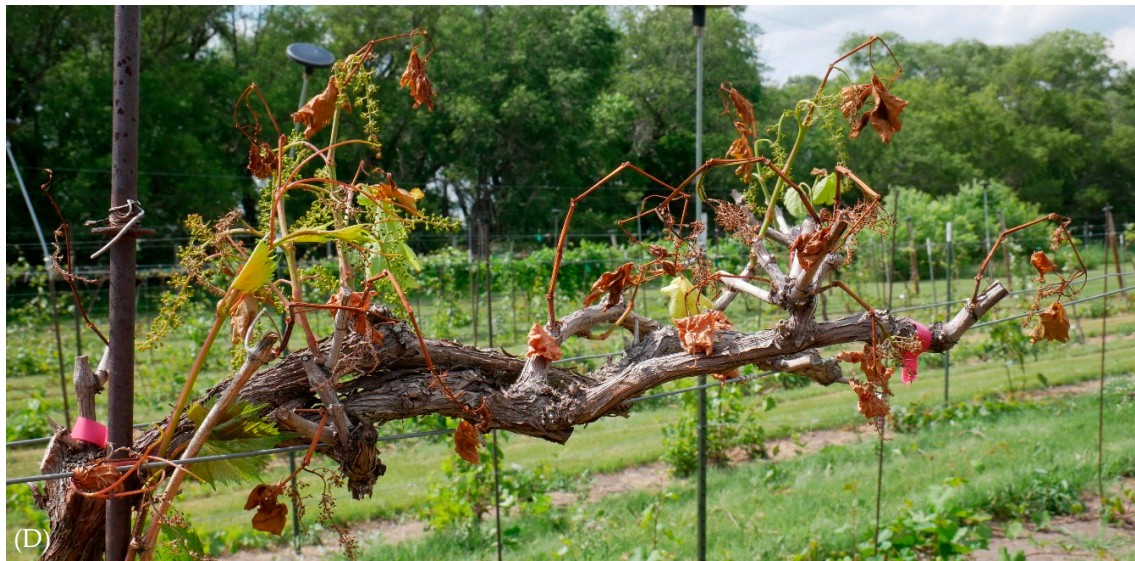

**Figure 2.** Trunk damage observed in the grape variety trial located at the NDSU Horticulture Research Farm near Absaraka, ND, USA during the late-spring to early-summer period of 2019. (**A**) Failure to break bud, 'Marquette' grapevines. (**B**) Trunk splitting of 'Frontenac' grapevines. (**C**) Trunk bleeding and subsequent microbial growth on two trunks of 'Frontenac gris' grapevines. (**D**) Trunk collapse shortly after bloom of "Frontenac" grapevines.

*2.3. Vineyard Re-Establishment*

Accounting for the catastrophic winter damage to the experimental vineyard, new trunks were retrained from actively growing suckers in the 2019 growing season (Figure 3). To manage vigor and reduce development of suckers as bull canes, the number of suckers per individual vine position within an experimental unit were manipulated for the development of new trunks with a target of two to four suckers per vine.

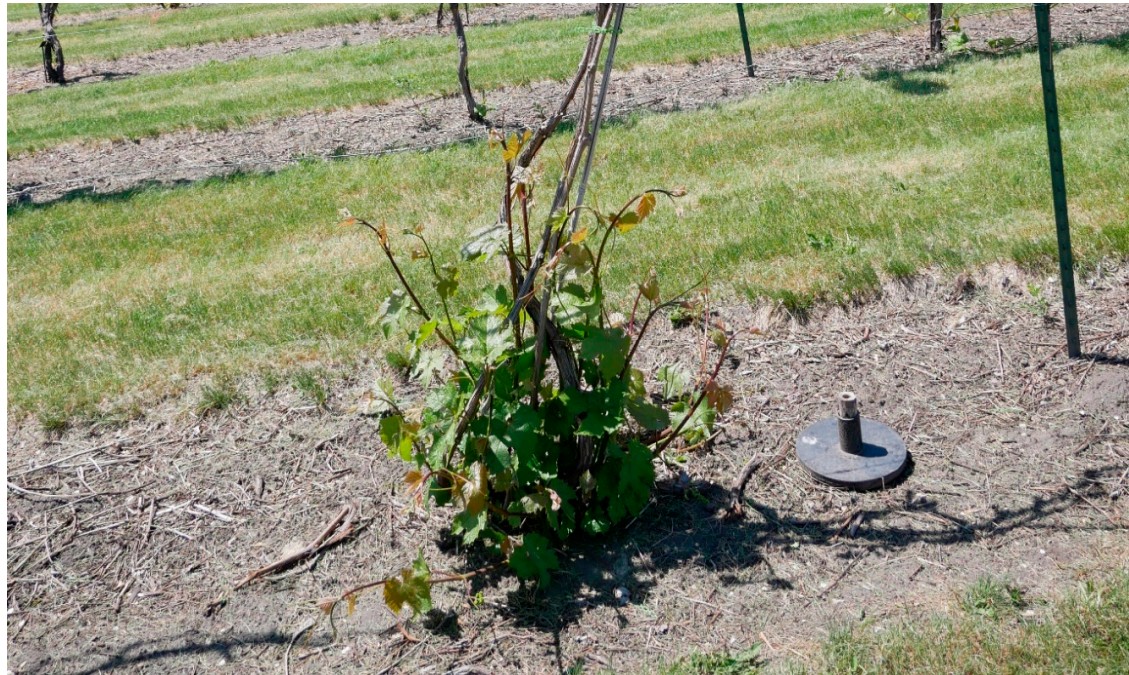

**Figure 3.** Suckers of "La Crescent" prior to thinning and training in the grape variety trial located at the NDSU Horticulture Research Farm near Absaraka, ND, USA.

To gain insight into the dynamics of trunk re-establishment following the harsh winter events, new suckers trained in 2019 were monitored for multiple traits at the end of the growing season. The traits measured in mid-October 2019, after leaf fall, included periderm development (node number and length) and sucker diameter (two measurements per sucker, large diameter and small diameter) at 25 cm above the soil surface. From these numbers, sucker circumference, sucker caliper ratio (larger diameter divided by smaller diameter for a given sucker), and internode length were calculated.

*2.4. Statistical Analysis*

Statistical analyses of sucker re-establishment metrics were conducted for replicate plots as a mixed-model analysis of variance (ANOVA) with genotype defined as a fixed effect and replicate defined as a random effect using JMP Pro 14.0.0 (SAS, Cary, NC). Visualization of DTA results was constructed using R statistical software version 3.6.1 within the ggplot2 package [18,19].

## 3. Results

*3.1. Differential Thermal Analysis of Dormant Buds*

Evaluation of bud cold-hardiness as measured by $LTE_{50}$ values indicated all tested cultivars were likely susceptible to damage from extreme winter events occurring in late Jan. (Figure 4). Early testing on 13 Nov. indicated 'Valiant' was the hardiest genotype (−29.2 °C) and 'Verona' the least hardy (−21.6 °C) with the remaining individuals lying in the narrow range from −23.8 °C ('St. Croix') to −25.9 °C ('Frontenac'). $LTE_{50}$ values coalesced as the season continued, although 'Valiant' remained the hardiest on 16 Dec. (−28.8 °C), while the remaining individuals' dormant bud $LTE_{50}$ values stayed between −25.9 °C and −27.2 °C. On the final testing date, $LTE_{50}$ values ranged from −26.7 °C to −32.5 °C, with 'Valiant' having the overall lowest $LTE_{50}$ value. However, 4 days later, grapevines were exposed to the season's lowest temperatures, reaching −36.8 °C on 30 January. Following this late January freeze event, two additional testing evaluations were conducted in February. However, nearly all cultivars failed to produce peaks, indicating a potentially high rate of bud mortality or substantial dehydration, reducing the capacity to observe clear peaks. Subsequent DTA testing was not conducted through the remainder of the dormant season (March–early May) in 2019.

*3.2. Winter Injury*

Early season evaluation revealed that 17 of 35 replicated genotypes failed entirely to break bud by June 2019. Genotypes "Alpenglow", E.S. 5-4-71, "GR 7", "Léon Millot", "Maréchal Foch", "Marquette", MN1220, MN1258, "Petite Amie", "Somerset Seedless", "Verona", "Itasca", and "Petite Pearl" were all rated as experiencing 100% trunk loss (Table 1). Contrastingly, at this stage, "Baltica", "Bluebell", "John Viola", "King of the North", and "Valiant" experienced the least trunk injury. Although "Frontenac", "Frontenac gris", and "Kay Gay" showed relatively low injury early in the growing season, they exhibited between 67% and 76% in-season trunk collapse. Approximately 25 vines out of a total of 578 experimental vines (across replicated and unreplicated plots) exhibited sap flow and microbial growth on trunk wounds prior to collapse; however, no consistent trend was observed for genotype or replicate location in field (data not shown). Overall, the top performing genotypes retaining a high rate of surviving trunks at the growing seasons' conclusion were 'Bluebell' (54%), "Baltica" (68%), 'John Viola' (89%), "King of the North" (93%), and "Valiant" (100%).

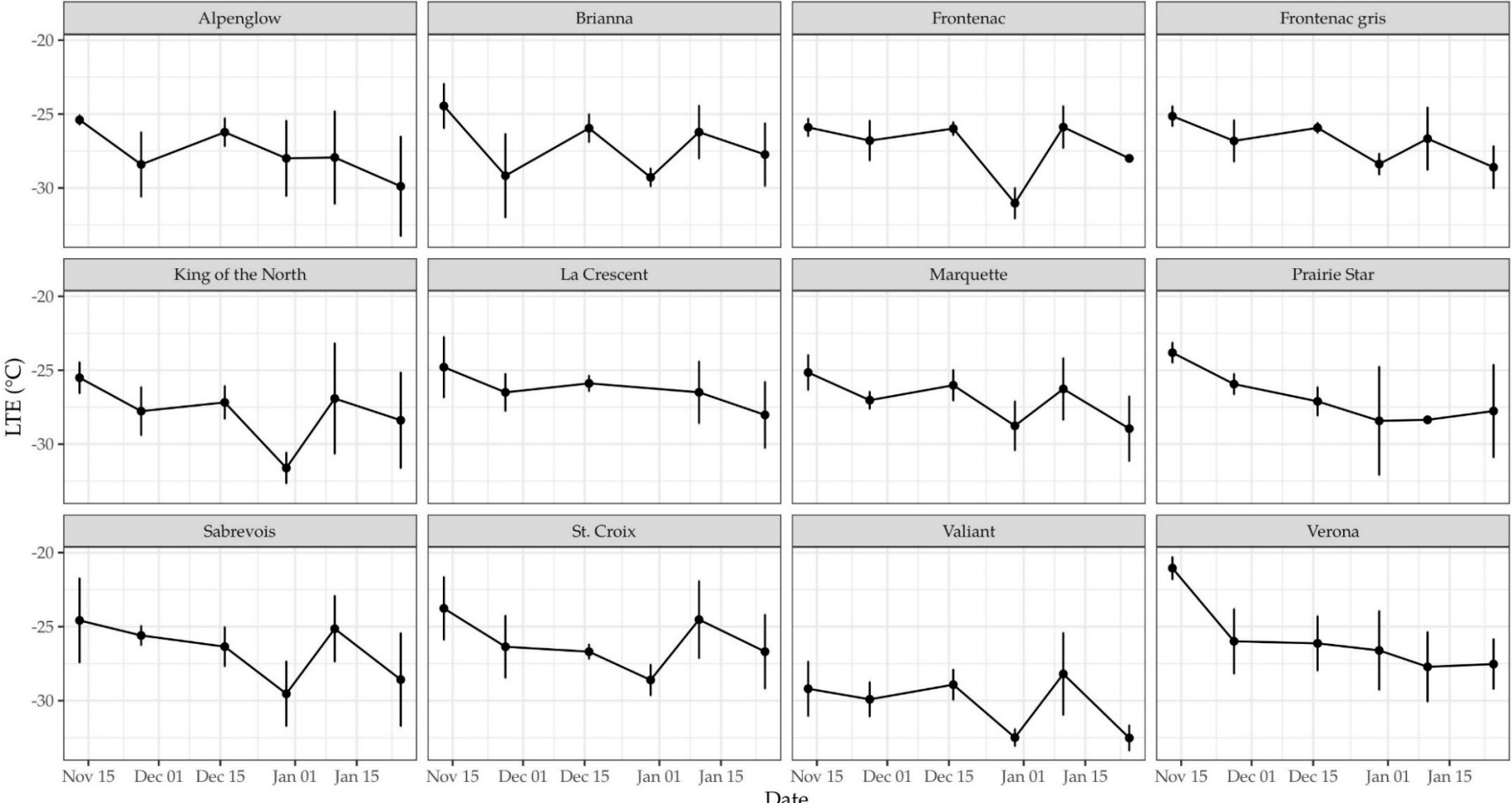

**Figure 4.** Mean LTE$_{50}$ values (± standard deviation) for 12 cultivars sampled during 2018–2019 winter in the grape variety trial located at the NDSU Horticulture Research Farm near Absaraka, ND, USA.

**Table 1.** Trunk survival evaluations for cold-hardy grapevine genotypes at the NDSU Horticulture Research Farm near Absaraka, ND, USA following the 2018–2019 winter.

| Genotype | Establishment Year | Percent Trunks Failing to Break Bud | Percent Trunks Collapsed Midseason | Percent Surviving Trunks |
|---|---|---|---|---|
| | | —Established Genotypes— | | |
| Alpenglow | 2004 | 100 | 0 | 0 |
| Baltica | 2004 | 14 | 18 | 68 |
| Bluebell | 2004 | 27 | 19 | 54 |
| Brianna | 2008 | 87 | 13 | 0 |
| Edelweiss | 2009 | 87 | 13 | 0 |
| E.S. 5-4-71 | 2004 | 100 | 0 | 0 |
| Frontenac | 2004 | 17 | 76 | 7 |
| Frontenac gris | 2004 | 29 | 67 | 4 |
| GR 7 | 2009 | 100 | 0 | 0 |
| John Viola | 2009 | 7 | 4 | 89 |
| Kay Gray | 2009 | 20 | 72 | 8 |
| King of the North | 2004 | 0 | 7 | 93 |
| La Crescent | 2004 | 94 | 6 | 0 |
| Laura's Laughter | 2009 | 92 | 8 | 0 |
| Léon Millot | 2009 | 100 | 0 | 0 |
| Louise Swenson | 2009 | 90 | 10 | 0 |
| Maréchal Foch | 2009 | 100 | 0 | 0 |
| Marquette | 2005 | 100 | 0 | 0 |
| MN1131 | 2004 | 42 | 29 | 29 |
| MN1200 | 2004 | 90 | 3 | 7 |
| MN1220 | 2008 | 100 | 0 | 0 |
| MN1235 | 2008 | 81 | 19 | 0 |
| MN1258 | 2008 | 100 | 0 | 0 |
| Petite Amie | 2008 | 100 | 0 | 0 |
| Prairie Star | 2008 | 96 | 4 | 0 |
| Sabrevois | 2004 | 78 | 18 | 4 |
| Somerset Seedless | 2004 | 100 | 0 | 0 |
| St. Croix | 2004 | 89 | 11 | 0 |
| Valiant | 2004 | 0 | 0 | 100 |
| Verona | 2007 | 100 | 0 | 0 |
| | | —Establishing Genotypes— | | |
| Adalmiina | 2017 | 88 | 12 | 0 |
| Crimson Pearl | 2014 | 92 | 0 | 8 |
| Frontenac blanc | 2012 | 89 | 11 | 0 |
| Itasca | 2017 | 100 | 0 | 0 |
| Petite Pearl | 2012 | 100 | 0 | 0 |

While not replicated, end-row unique genotypes composed of two vine panels were examined for survival to gain insight into their suitability for regional production or as parental material in future crosses. The only genotypes with surviving trunks at the conclusion of the 2019 growing season were E.S. 9-4-72, E.S. 10-18-58, and JB 13 (Table A1). The highest rate of trunk loss at the start of the season was observed for E.S. 3–20–33, E.S. 4–22–60, E.S. 10–18–14, and E.S. 10-18–75; for these genotypes, all trunks (spurs, canes, and shoots above soil) failed to break bud at the onset of the 2019 season.

*3.3. Vineyard Re-Establishment*

Retained sucker number was consistently within the target range of two to four suckers (Table 2). The genotypes with the highest retrained number of suckers per individual plant were "Alpenglow", "La Crescent" and "Sabrevois", while "Leon Millot" and "Edelweiss" had the lowest number of retained suckers. The length of the retained live suckers exceeded 200 cm for all genotypes except "Louise Swenson". "La Crescent" had the longest mean sucker length (360.8 cm), while "Louise Swenson" had the smallest sucker lengths (148.4 cm). This comparison was carried on in their sucker node number, as "Louise Swenson" again produced the lowest number of nodes (14.5) and "La Crescent" produced the most (26.4).

**Table 2.** Sucker metrics for re-established grapevine suckers trained in 2019 after winter kill during the 2018–2019 winter at the NDSU Horticulture Research Farm near Absaraka, ND, USA.

| Genotype | Retained Suckers (No.) | | Live Nodes (No./Sucker) | | Live Sucker Length (cm/Sucker) | | Internode Length (cm) | | Sucker Circumference (mm) | | Sucker Caliper Ratio | |
|---|---|---|---|---|---|---|---|---|---|---|---|---|
| Alpenglow | 3.7 | a [1] | 20.8 | bcde | 246.8 | bc | 11.6 | bcde | 22.3 | efg | 1.12 | abc |
| Brianna | 2.4 | ab | 21.0 | abcde | 263.4 | bc | 12.5 | abc | 25.6 | cdef | 1.09 | bc |
| Edelweiss | 2.1 | b | 19.6 | bcdef | 225.2 | bcd | 11.3 | bcdef | 23.2 | defg | 1.12 | abc |
| E.S. 5-4-71 | 2.7 | ab | 18.6 | cdef | 232.4 | bcd | 12.5 | abcd | 24.6 | cdefg | 1.10 | abc |
| Frontenac | 3.0 | ab | 18.7 | def | 230.0 | bcd | 11.7 | bcde | 29.5 | bc | 1.11 | abc |
| Frontenac gris | 3.1 | ab | 23.1 | abcd | 273.6 | b | 11.7 | bcde | 30.9 | ab | 1.11 | abc |
| GR 7 | 2.9 | ab | 24.5 | abc | 258.2 | bc | 10.2 | efg | 22.3 | efg | 1.12 | abc |
| Kay Gray | 2.7 | ab | 21.1 | abcde | 228.1 | bcd | 10.7 | cdefg | 22.3 | efg | 1.12 | abc |
| La Crescent | 3.8 | a | 26.4 | a | 360.8 | a | 13.9 | a | 34.4 | a | 1.13 | abc |
| Laura's Laughter | 2.5 | ab | 22.2 | abcd | 252.6 | bc | 11.3 | bcdef | 26.6 | bcde | 1.11 | abc |
| Léon Millot | 2.1 | b | 26.2 | ab | 255.1 | bc | 9.7 | fgh | 23.8 | defg | 1.12 | abc |
| Louise Swenson | 2.3 | ab | 14.5 | f | 148.4 | d | 10.1 | efg | 21.1 | efg | 1.07 | c |
| Maréchal Foch | 2.6 | ab | 24.4 | abcd | 196.1 | cd | 8.0 | h | 21.2 | fg | 1.15 | ab |
| Marquette | 3.1 | ab | 21.8 | cd | 204.6 | bcd | 9.2 | gh | 23.4 | efg | 1.10 | abc |
| Prairie Star | 3.2 | ab | 19.1 | cdef | 196.4 | bcd | 10.2 | efg | 19.6 | g | 1.11 | abc |
| Sabrevois | 3.7 | a | 20.3 | bcdef | 246.5 | bc | 11.9 | bcde | 28.6 | bcd | 1.11 | abc |
| Somerset Seedless | 2.9 | ab | 20.6 | bcdef | 225.0 | bcd | 10.6 | cdefg | 23.0 | efg | 1.16 | a |
| St. Croix | 3.2 | ab | 15.8 | ef | 205.8 | bcd | 12.7 | ab | 24.6 | def | 1.08 | c |
| Verona | 2.6 | ab | 23.9 | abcd | 251.7 | bc | 10.5 | defg | 25.8 | cdef | 1.13 | abc |
| F Ratio | 2.58 | | 7.47 | | 8.02 | | 14.72 | | 15.38 | | 2.40 | |
| *p* | 0.0006 | | <0.0001 | | <0.0001 | | <0.0001 | | <0.0001 | | 0.0009 | |

[1] Means followed by the same letter within columns are not significantly different according to Tukey's HSD at $\alpha = 0.05$.

Retained sucker internode length was smallest for 'Maréchal Foch' at 8.0 cm and largest for 'La Crescent' at 13.4 cm (Table 2). Only "Léon Millot", "Maréchal Foch", and "Marquette" had internode lengths below 10.0 cm. In contrast, "Brianna", "La Crescent", and "St. Croix" were the only individuals with internode lengths exceeding 12.0 cm.

Sucker circumference was greatest for "La Crescent" (34.4 mm), "Frontenac gris" (30.9 mm) and "Frontenac" (29.5 mm) genotypes, and smallest for "Prairie Star" (19.6 mm) (Table 2). "Louise Swenson" (1.07) and "St. Croix" (1.08) suckers had the lowest sucker caliper ratio. They were only statistically separable from 'Somerset Seedless' and "Maréchal Foch".

Separation of sucker metrics for re-established grapevines was less frequently observed among the more recently planted genotypes (Table 3). "Adalmiina" trunks had the lowest number of live nodes and subsequently the shortest length. 'Petite Pearl' trunks were the longest with the greatest internode length.

For the non-replicated genotypes, live sucker nodes and lengths ranged from 12.8 to 30.0 nodes and 137.8 to 408.9 cm (Table A2). Only three genotypes failed to produce 20 nodes, E.S. 3-20-33, E.S. 10–18-14, and LxT 10. E.S. 3–20–33 (137.8 cm) and LxT 10 (139.3 cm) also produced the smallest sucker lengths while DM 8521 (322.6 cm), E.S. 10-18-75 (368.9 cm), and E.S. 9-4-72 (408.9 cm) had the longest suckers. Internode lengths were greater than 10.0 cm for all but LxT 10, and sucker circumference was only below 20.0 mm for LxT 10 and E.S. 3-20-33.

**Table 3.** Sucker metrics for re-established grapevine suckers of establishing genotypes trained in 2019 after winter kill during the 2018–2019 winter at the NDSU Horticulture Research Farm near Absaraka, ND, USA.

| Genotype | Retained Suckers (No.) | | Live Nodes (No./Sucker) | | Live Sucker Length (cm/Sucker) | | Internode Length (cm) | | Sucker Circumference (mm) | | Sucker Caliper Ratio | |
|---|---|---|---|---|---|---|---|---|---|---|---|---|
| Adalmiina | 2.5 | ns [1] | 17.3 | b | 124.7 | b | 7.3 | b | 19.7 | ns | 1.06 | ns |
| Crimson Pearl | 3.0 | | 24.1 | a | 198.5 | ab | 7.7 | b | 23.4 | | 1.13 | |
| Frontenac blanc | 2.4 | | 24.5 | a | 178.7 | ab | 7.0 | b | 22.4 | | 1.15 | |
| Itasca | 2.9 | | 22.5 | a | 177.3 | ab | 7.8 | b | 28.0 | | 1.13 | |
| Petite Pearl | 2.2 | | 23.5 | a | 226.5 | a | 9.5 | a | 23.0 | | 1.10 | |
| F Ratio | 1.14 | | 5.17 | | 5.94 | | 7.45 | | 1.64 | | 1.50 | |
| *p* | 0.346 | | 0.0006 | | 0.0002 | | <0.0001 | | 0.1679 | | 0.2055 | |

[1] Means followed by the same letter within columns are not significantly different according to Tukey's HSD at $\alpha = 0.05$; ns = not significant.

## 4. Discussion

During the 2018–2019 winter, nearly all grapevine trunks were injured. Among the genotypes tested, "Valiant" consistently performed as the most cold-injury-resistant according to $LTE_{50}$ values for primary buds. This was supported by its relative survival throughout the 2019 growing season. Despite the inconsistent and less cold-hardy $LTE_{50}$ values of "King of the North" relative to "Valiant", it emerged in the spring largely unscathed based on trunk survival observations.

The results indicate that the 2018–2019 winter was not only damaging to grapevine dormant buds, but also the important conductive tissues. Ice crystal formation transmits tensile stress within grapevine trunks to deform and damage xylem and phloem tissues [20]. The rate and extent of this damage depends on the water, carbohydrate, and protein contents. Moderately injured conductive tissue may be regenerated by cambium-produced callus, which subsequently differentiates, yielding new phloem and xylem [21]. Previous work has demonstrated that phloem damage may have no observed impact on budbreak, growth, yield, or fruit quality of grapevines [22]. Thus, grapevines may be apt to re-establish new phloem tissue under production settings, given the presence of live, healthy buds. Contrastingly, severe phloem and cambium damage reduces survival and yield components of grapevines [23]. Within evaluated, North Dakota-grown, surviving genotypes there was likely damage to conductive tissue. However, the presence of live buds may have enabled some genotypes

to regenerate damaged phloem tissue, while other genotypes experiencing full bud death or more extensive phloem and cambial damage may have been less capable of conductive tissue regeneration.

Following budbreak, bleeding of xylem exudates coupled with microbial growth on this nutrient-rich fluid was observed from multiple genotypes' trunks. In addition, massive trunk cavitation was also observed where trunks and cordons were split open. In all instances within the grapevine variety trial and nearby experimental plot of "Prairie Star" grapevines, observed trunk bleeding with microbial growth was followed by canopy collapse by mid-season during 2019. The extensive trunk injury observed (bleeding, splitting, and collapse) may be indicative of the need for extension of field phenotyping protocols beyond bud characterization, incorporating evaluations of phloem and xylem as supplementary metrics to decipher what variances exist in genotypic response to extreme cold-weather.

Seasonal temperature fluctuations, especially near the base of grapevine trunks, may have predisposed grapevine tissue to injury due to changes in acclimation status [24]. Daily minimum temperatures have been demonstrated as driving forces for shifts in cold-hardiness. With the relatively low daily minimum temperatures experienced going into major freeze events, the genotypes examined in North Dakota may have achieved the greatest extent of cold-hardiness that they are physiologically capable of obtaining. Although, Zabadal et al. [5] noted critical lethal temperatures as −26 to −29 °C for hardy *V. labrusca*-types (Concord), and −29 to −34 °C for the very hardy *V. riparia*-based hybrids grapes ('Frontenac' and 'La Crescent'); in this study, the highest survival was observed for hybrid grapevines containing both *V. riparia*- and *V. labrusca*-derived characteristics. The five genotypes exceeding 50% trunk survival, ("Valiant", "King of the North", "Baltica", "Bluebell", and "John Viola") are grapevines producing red to noir grapes with *V. labrusca* background. Their heavily *V. labrusca* aromas make them potentially ideal juice and jam grapevines, but reduce their immediate appeal for winemaking potential. Interestingly, despite the presence of public and private breeding efforts to develop cold-hardy wine grapevines, two individuals with high rates of trunk survival, "King of the North" and "John Viola" are chance seedlings, likely with extensive *V. labrusca* and *V. riparia* background in their undocumented parentage.

Most of North Dakota is within the USDA plant hardiness zones of 4a and 3b [25], which extends beyond the target breeding environment of Elmer Swenson who bred grapevines in Osceola, WI [26]. Thus, it is understandable that no Elmer Swenson produced varieties exceeded 50% trunk survival in the main variety trial. 'Kay Gray' and 'Sabrevois' performed the best, retaining 8 and 4% of their trunks, respectively. Unreplicated individuals, E.S. 9-4-72 and E.S. 10-18-58 retained 33% of their trunks. These are both juice and jam type noir grapevines from Elmer Swenson's breeding efforts with characteristic *V. labrusca* aromas.

"La Crescent" produced the longest suckers with the most nodes. It also had the greatest sucker circumference and longest internodes, despite having the most retained suckers. This propensity to produce thick wood with large internodes may contribute to the instability of "La Crescent" for grape growers in North Dakota. Thick wood has been shown to be less cold-hardy when examining "Cabernet Sauvignon"; bull-canes had larger surface area of pith, xylem, and phloem; larger xylem and phloem vascular transport units (VTUs); and more VTUs [27]. The increased risk of freeze damage to large canes indicates the need to avoid bull cane development when retraining trunks. In attempts to combat vigor within Ohio grown 'Cabernet Sauvignon', researchers retained an average of 10 shoots per vine while transitioning to a fan training system, yet 60% of shoots were reported as large. Their results indicate that the retention of more shoots in trunk re-establishment may not be sufficient to devigorate grapevines; shoot number alone may not acceptably dictate cane diameter nor a vine's tendency for bull cane production.

To alter genotypes' cane and trunk morphology towards the goal of reducing grapevine vigor, chemical growth regulator applications to renewal trunks may be investigated. Use of cultural practices reducing internode length of canes, vigor of laterals, and total leaf area may contribute to the reduction in individual vine vigor [28,29]. These practices may subsequently reduce the propensity of treated vines to produce bull canes bearing large xylem, phloem, and expansive VTUs. As such, these treatments may be useful for growers targeting the goal of increased relative hardiness of conductive tissues due to their

reduced cellular size. Techniques to inhibit vigor of grapevines, such as under-vine cover cropping, via reducing available water and nutrients may also aid in slowing overly vigorous genotypes [30].

While the 2018–2019 winter served as an extreme anomaly, it is worth considering whether genotypes may be deemed suitable for North Dakota conditions if the entirety of their trunks are susceptible to winter-kill. If these genotypes cannot be stimulated to increase their cold-hardiness through either chemical or management practices, they may not be viable for regional production. Under the extreme constraints of North Dakota, however, it may be worthwhile to consider some of these genotypes with high vigor and low winter survival because some of the growth characteristics, which compel them to be highly cold-susceptible, may also contribute to their relatively rapid return to productivity, enhancing their capacity to swiftly re-establish trunks and reoccupy canopy space.

Yield loss can be attributed to phloem damage because of freezing temperatures [23]. These winter freeze events, which alter the grapevine phloem, can reduce vigor (shorter, weaker shoots, with smaller, less expanded leaves), berry number per cluster, and lead to delayed grapevine phenology [23,24]. Resulting degeneration of phloem is interrelated with reduced cambial activity under field conditions; this may be implicated in long-term accumulated damage to grapevine permanent tissue such as cordons and trunks [23]. This "hidden" damage to perennial tissue threatens grapevine longevity, especially under demanding annual winter conditions.

In re-establishing cordons of 'Merlot' following grapevine decline after multiple winter freeze events, Gohil et al. [31] noted increased uniformity when entire cordons were removed. However, this drastic method was coupled with a multi-year delay before yield returned to commercially acceptable levels. The economic impact of yield loss associated with retraining permanent grapevine structures is a critical choice for producers, and under the harsh reality of complete vineyard retraining, it may be advisable for regional producers to maintain a state of constant-flux in the form of constant vineyard renewal. Differentially aged wood (cordons and trunks) would have a diversified history of freeze damage, and subsequently they may acclimate, experience, repair, and respond to major freeze events distinctly differently. In selecting new suckers, beyond avoiding bull-wood, growers must account for greater cane tissue and primary bud survival rate with well exposed, darkly colored wood, and should be careful when retraining vines to reduce over-crowding of wood and leaf tissue [32].

Other major considerations for producers going forward include the need for evaluation of different trellis and training systems, which may offer greater winter protection potential than a standard single high-wire bilateral cordon trellis-training system. Likewise, supplemental winter protection may be necessary via soil/snow burial, or through use of geotextile fabrics, if consistent production of non-adapted genotypes is desirable for producers [33,34]. Training systems worth considering include fan, dragon, crawled cordon, low cordon, mini-J, or head pruning near the soil-surface to facilitate burial or supplemental protection of fruiting wood against dangerous winter temperatures [13,35,36]. Ultimately, an emphasis must be placed on multi-trunk training systems to ensure rapid replacement of dead permanent structures [37]. For consistent yields and viability, grapevine growers in North Dakota may need to adapt management practices geared towards the protection of cold-hardy grapevine varieties in order to mimic the production of cold-tender *V. vinifera* within marginal climates.

**Author Contributions:** Conceptualization, A.S., B.K., J.S., and H.H.-V.; methodology, A.S., B.K.; formal analysis, A.S.; investigation, A.S., B.K., J.S., C.A.; data curation, A.S.; writing—original draft preparation, A.S., B.K.; writing—review and editing, A.S., B.K., J.S., H.H.-V.; visualization, A.S.; supervision, H.H.-V. All authors have read and agreed to the published version of the manuscript.

**Funding:** This research received no external funding.

**Acknowledgments:** The authors would like to thank Matthew Brooke, Nickolas Theisen, Binu Rana, and Ryan Archer for their assistance.

**Conflicts of Interest:** The authors declare no conflict of interest.

## Appendix A

**Table A1.** Trunk survival of unreplicated cold-hardy grapevine genotypes (two-vine plot/genotype) at the NDSU Horticulture Research Farm near Absaraka, ND, USA following the 2018–2019 winter.

| Genotype | Percent Trunks Failing to Break Bud | Percent Trunks Collapsed Midseason | Percent Surviving Trunks |
|---|---|---|---|
| DM 8521 | 0 | 100 | 0 |
| E.S. 3-20-33 | 100 | 0 | 0 |
| E.S. 4-22-60 | 100 | 0 | 0 |
| E.S. 9-4-72 | 50 | 17 | 33 |
| E.S. 10-18-14 | 100 | 0 | 0 |
| E.S. 10-18-58 | 0 | 67 | 33 |
| E.S. 10-18-75 | 100 | 0 | 0 |
| JB 13 | 33 | 33 | 33 |
| LxT 10 | 67 | 33 | 0 |

**Table A2.** Sucker metrics for unreplicated cold-hardy grapevine genotypes (two-vine plot/genotype) trained during the 2019 growing season at the NDSU Horticulture Research Farm near Absaraka, ND, USA.

| Genotype | Retained Suckers (No.) | Live Nodes (No./Sucker) | Live Sucker Length (cm/Sucker) | Internode Length (cm) | Sucker Circumference (mm) | Sucker Caliper Ratio |
|---|---|---|---|---|---|---|
| DM 8521 | 1.0 | 30.0 | 322.6 | 10.7 | 37.9 | 1.21 |
| E.S. 3-20-33 | 2.0 | 12.8 | 137.8 | 10.8 | 19.0 | 1.21 |
| E.S. 4-22-60 | 3.5 | 21.1 | 238.0 | 10.7 | 32.6 | 1.09 |
| E.S. 9-4-72 | 1.5 | 27.7 | 408.9 | 14.6 | 32.4 | 1.06 |
| E.S. 10-18-14 | 3.0 | 19.0 | 210.0 | 10.4 | 23.1 | 1.12 |
| E.S. 10-18-58 | 2.0 | 21.3 | 269.9 | 11.5 | 29.3 | 1.10 |
| E.S. 10-18-75 | 2.0 | 29.5 | 368.9 | 12.3 | 30.1 | 1.19 |
| JB 13 | 3.0 | 24.0 | 256.5 | 10.7 | 25.0 | 1.12 |
| LxT 10 | 3.0 | 13.5 | 139.3 | 8.7 | 18.0 | 1.18 |

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
