# Peer review of "Cold-Hardy Grape Cultivar Winter Injury and Trunk Re-Establishment Following Severe Weather Events in North Dakota"

_horticulturae, doi:10.3390/horticulturae6040075_

Round 1

Reviewer 1 Report

This paper provides one year of data on grapevine cold-hardiness and recovery in response to a major cold weather event. The researchers had set out to collect LTE of key varieties and accessions but, severe weather changed the experiment. I applaud these researchers for taking advantage of a "test winter" and persevering through data collection on the plant materials. Overall, this paper is well written and would be very difficult to have multi-year data and thus provides a unique opportunity present and extreme case of winter injury. I have provided suggested changes/edits by line number for the authors to consider and/or correct. 

17: space between 37 and degree, this is the way it is reported in text elsehwere

27: "genotypes", although accurate this use of genotype through out the paper could be changed to "varieties, accessions, germplasm, etc." My personal preference is to limit the use of "genotype" to when it refers to genotypic information such as genetic markers or sequences.

33: cold temperatures are major limitation to grapevine production everywhere with cold temperatures. I'm not sure what is meant by "continental". I would say Eastern US if you are really interested in discussing non west-coast grapevines

44: It is prudent that the reader know that these vines are own-rooted, so that suckers can be trained; this will be different than most grape production regions around the world.

51: or straw!!

54: "and grapevine" seems redundant

61: consider alternative for "genotype"

62-63: sentence is awkward with "per se" and "potential producers"; needs clarification. producer usually means grower/farmer in these contexts

65: clarify that this experiment is not multi year 2018 and 2019, but rather initiated in 2018 and continued through 2019 growing season

Figure 1: Not clear what "normal" means. i'm guessing this is an average over ## of years, give that somewhere (in figure legend or caption)

81: "bi weekly" is troublesome word as in indicates 2x per week or every other week. Clarify which is meant

94: I think this should read -4C hr -1

103-104 consider giving year "may 2019"

120: I have only seen use of "bull cane" not bull "wood"

124: encompassed gives connotations that the periderm has grown over the bud. You could check resources (Bob Pool, Andrew Reynolds) for other terms. "ripe periderm" may be appropriate

125: I do not understand the trunk caliper ratio: is the larger trunk the dead and older trunk? Please clarify here

134: Why was replicate as a random effect? I assume you have an unbalanced data set. This should be more clear. Table 1 would benefit from sample number (aka rep number) for the reader to know about the experimental design. It is not clear in the MM section how many reps of any plats were tested. 

Similarly, results are presented in lines 169-174 that aren't mentioned in the MM section, or if so was not enough to prepare the reader for these 'extra results'

142: "measured"

143: "prepared" anthropomorphizes grape vines

149: Does not need to start new paragraph

160: 17 of ## genotypes? it would be useful to know this quickly

Table 1: I think it is worth clarifying that failure to break bud includes trunks and cordons and canes. So the column header could be simplified and the text also clarified that buds were not detected along any of those structures. For consistency the (Percent %) is redunant with itself and in a different format than Table A1 (has lines). Both need to be corrected

180. Although 3.0 trunks is your statistical cut off, 3 makes more sense from a production stand point

194: not clear what "approaching round" refers to and 196: what flat means. This references back to the MM on trunk ratio

Table 2: foot note "comment" seems like wrong word, "indicates" 

Table 2: caption: "trunks" could be "suckers" here to be more accurate (and in column heads) and to distinguish between the dead trunks and new tissue.

Table 3: Immature has biological meaning (not producing fruit, juvenile) is this what is meant here? if not, "younger" should be used and also corrected in the MM. Same comments about the word "trunk" as above

Figure 5: The PCA plot doesn't contribute greatly to this project. The presentation of 3 dimensions is not too useful when 2 are only shown in the biplot. A simple correlation table may provide satisfactory analysis of these interactions.

236: "moderately hardy" is VERY relative here, comparing your 2 most cold-hardy individuals.

238: Check grammar "was not only been damaging"

247: check grammar and context. what is "surviving" mean. Its no clear to me that any plants fully died. 

270: I'm not sure the big emphasis on labrusca here, as these are clearly riparia too!

278: produced -> developed, bred

282: does it need to be in must? does it need to be at harvest? these seem like just "berry' attributes

287: "surpassing defintions" who's? what does this mean?

295: bull wood again

303: "recently developed growth-regulators" is that a technique? a product used on grapes? I think this sentence needs clarification

314: simply "yield loss"

317: "reduced-to-negligible" just "reduced"

323: "a delay before yield returned to acceptable levels"; this is awkward. How long of a delay? hours? years? Comparing Vinifera here seems out of touch with the types of grapes in this study. any evidence from hybrids in the literature?

326: "ever on-going" could just be "on-goin"

All references need to be checked. Principally, make sure to italicize scientific names. 

Author Response

This paper provides one year of data on grapevine cold-hardiness and recovery in response to a major cold weather event. The researchers had set out to collect LTE of key varieties and accessions but, severe weather changed the experiment. I applaud these researchers for taking advantage of a "test winter" and persevering through data collection on the plant materials. Overall, this paper is well written and would be very difficult to have multi-year data and thus provides a unique opportunity present and extreme case of winter injury. I have provided suggested changes/edits by line number for the authors to consider and/or correct. 

  • 17: space between 37 and degree, this is the way it is reported in text elsewhere

Response: Thank you, adjustment made.

  • 27: "genotypes", although accurate this use of genotype through out the paper could be changed to "varieties, accessions, germplasm, etc." My personal preference is to limit the use of "genotype" to when it refers to genotypic information such as genetic markers or sequences.

Response: To avoid monotony, we worked to replace the work “genotype” with cultivars in instances when all accessions evaluated were named cultivars. For the majority of the paper, however, we were obligated to stick with genotype, as it is the most accurate term to describe the diversity of material evaluated. From a molecular genetic perspective, we acknowledge the potential confusion from the use of the word genotype; however, from an applied production/breeding/ variety evaluation standpoint genotype appears to be the term most accurate at encompassing the material evaluated.

  • 33: cold temperatures are major limitation to grapevine production everywhere with cold temperatures. I'm not sure what is meant by "continental". I would say Eastern US if you are really interested in discussing non west-coast grapevines

Response: “continental” adjusted to “Eastern”.

  • 44: It is prudent that the reader know that these vines are own-rooted, so that suckers can be trained; this will be different than most grape production regions around the world.

Response: Agreed, clarification was required;

Adjustment to read: Dead trunks of own-rooted vines require renewal with new shoots, typically suckers, arising at or just below the soil level [10,11].

  • 51: or straw!!

Adjusted to include straw and geotextiles: ….. grapevines may be protected by covering with soil, snow, straw, or geotextile fabrics to reduce grapevine loss…..

  • 54: "and grapevine" seems redundant

Omitted redundant “and grapevine”: At this time, no commercial grapevine producers in North Dakota actively practice this method of trunk winter protection.

  • 61: consider alternative for "genotype"

Adjustment: “genotype” changed to “germplasm”; continued with this and/or to cultivar when logical throughout the paper to decrease monotony.

  • 62-63: sentence is awkward with "per se" and "potential producers"; needs clarification. producer usually means grower/farmer in these contexts

Removed redundant “potential producers”

Adjusted: Further, goals included the identification of genotypes with potential to perform well for local farmers or for use as parents in regional grapevine breeding efforts.

  • 65: clarify that this experiment is not multi year 2018 and 2019, but rather initiated in 2018 and continued through 2019 growing season

Clarified through addition of planting description paragraph:

In 2004, a cold-hardy hybrid grapevine variety trial was established at the NDSU HRF, near Absaraka, ND, USA. From 2004 to 2009, nearly 40 own-rooted genotypes (both released cultivars and breeding germplasm) were planted for evaluation; however, only 30 were retained through the initiation of this experiment in 2018. Between 2012 and 2017, additional own-rooted genotypes were planted for characterization in rows adjacent to the earlier established (2004-2009) variety trial planting; of these newly planted genotypes, five recently released cultivars were evaluated in this experiment. The entire unirrigated experimental vineyard plot was planted on a 0 to 2 percent slope in Warsing, sandy loam soil. Vines were trained to multiple trunks with spur-pruned, bilateral cordons on a single high-wire at 1.8 m above the soil level. Rows 83 m long were established in a North to South orientation with a spacing of 2.4 m between vines in-row and 3.0 m between rows. A 0.5 m weed-free strip was maintained below the vine rows using tillage combined with pre-emerge (Flumioxazin, Chateau ®, Valent USA, San Ramon, CA, USA) and post-emerge (Glufosinate, Rely ®, BASF, Florham Park, New Jersey, USA) herbicide applications. Vineyard mid-rows were planted with red fescue (Festuca rubra). All grapevines genotypes were planted into four-vine experimental units in a randomized complete block-design with four replicates clustered by period of planting. A total of 16 vines were examined per replicated genotype; an additional nine unreplicated genotypes (two vines per plot) were also examined for trunk survival and retraining metrics. Evaluations of grapevine cold-hardiness were conducted through the 2018-2019 dormant season, and evaluations of grapevine trunk survival and retraining were conducted in the 2019 growing season.”

  • Figure 1: Not clear what "normal" means. i'm guessing this is an average over ## of years, give that somewhere (in figure legend or caption)

Clarified: Figure 1. Normal (1971-2000) daily temperatures and temperatures from September 2018 to May 2019 recorded at the North Dakota Agriculture Weather Network weather station near Prosper, ND USA.

  • 81: "bi weekly" is troublesome word as in indicates 2x per week or every other week. Clarify which is meant

Clarified: Dormant canes were collected twice monthly for a subset of twelve cultivars from early Nov. 2018 through mid Feb. 2019.

  • 94: I think this should read -4C hr -1

Adjustment: added – symbol to re-emphasize the decrease in temperature.

  • 103-104 consider giving year "may 2019"

Adjustment: May, July, November now all followed by year (2019)

  • 120: I have only seen use of "bull cane" not bull "wood"

Adjusted to read: To manage vigor and reduce development of trunks as bull canes, the number of suckers per individual vine position were manipulated for the development of new trunks with a target of two to four suckers per vine.

Note: Altered to bull cane throughout.

  • 124: encompassed gives connotations that the periderm has grown over the bud. You could check resources (Bob Pool, Andrew Reynolds) for other terms. "ripe periderm" may be appropriate

Adjusted: Removed “encompassed” from text.

  • 125: I do not understand the trunk caliper ratio: is the larger trunk the dead and older trunk? Please clarify here

Added clarification in M&M: The traits measured in mid-October 2019, after leaf fall, included periderm development (node number and length), and sucker diameter (two measurements per sucker, large diameter and small diameter) at 25 cm above the soil surface. From these numbers, sucker circumference, sucker caliper ratio (larger diameter divided by smaller diameter for a given sucker), and internode length were calculated.

  • 134: Why was replicate as a random effect? I assume you have an unbalanced data set. This should be more clear. Table 1 would benefit from sample number (aka rep number) for the reader to know about the experimental design. It is not clear in the MM section how many reps of any plats were tested. 

Similarly, results are presented in lines 169-174 that aren't mentioned in the MM section, or if so was not enough to prepare the reader for these 'extra results'

Adjusted: M&M Section enhanced to include description of replication, experimental plot, and vine maintenance and to account for unreplicated genotypes prior to includion in anecdotal evaluation.

  • 142: "measured"

Adjusted to “measured”

  • 143: "prepared" anthropomorphizes grape vines

Adjusted: Evaluation of bud cold-hardiness as measured by LTE50 values indicated all cultivars were likely susceptible to damage from extreme winter events occurring in late Jan. (Fig. 4).

  • 149: Does not need to start new paragraph

Adjustment: paragraph break removed.

  • 160: 17 of ## genotypes? it would be useful to know this quickly

Adjustment: “….17 of 35 replicated genotypes…..”

  • Table 1: I think it is worth clarifying that failure to break bud includes trunks and cordons and canes. So the column header could be simplified and the text also clarified that buds were not detected along any of those structures. For consistency the (Percent %) is redunant with itself and in a different format than Table A1 (has lines). Both need to be corrected

Adjustment: Removed %, added details in M&M to define the trunks that failed to break bud-

“The total number of trunks per experimental unit were recorded and percentages of trunks that failed to break bud from any point (2019 shoot growth entirely absent from spurs, canes, cordons, and the length of the trunk to the soil level), collapsed mid-season (green shoots appeared early in season, but ceased growth), or survived through the entirety of the 2019 growing season were calculated.”

  • Although 3.0 trunks is your statistical cut off, 3 makes more sense from a production stand point

Adjustment: “Retained trunk number was consistently within the target range of two to four trunks”

  • 194: not clear what "approaching round" refers to and 196: what flat means. This references back to the MM on trunk ratio

Adjustment: altered M&M as above, removed “approaching round”/flat discussion.

  • Table 2: foot note "comment" seems like wrong word, "indicates" 

Adjustment: 1Means followed by the same letter within columns are not significantly different according to Tukey’s HSD at α =0.05

  • Table 2: caption: "trunks" could be "suckers" here to be more accurate (and in column heads) and to distinguish between the dead trunks and new tissue.

Adjustment: altered trunk to suckers here (and throughout the manuscripts) to represent new growth

  • Table 3: Immature has biological meaning (not producing fruit, juvenile) is this what is meant here? if not, "younger" should be used and also corrected in the MM. Same comments about the word "trunk" as above

Response: Generally, immature as juvenile accurately describes the status of the younger cultivars with no history of yield at the site; however, due to the lack of clarity from the term, altered to make more clear.

Adjustment: “immature” shifted to “establishing”

  • Figure 5: The PCA plot doesn't contribute greatly to this project. The presentation of 3 dimensions is not too useful when 2 are only shown in the biplot. A simple correlation table may provide satisfactory analysis of these interactions.

Adjustment: Agreed, removed PCA plot and results presentation as they do little to expand on the results of the study.

  • 236: "moderately hardy" is VERY relative here, comparing your 2 most cold-hardy individuals.

Clarified: “Despite the inconsistent and less cold-hardy LTE50 values of ‘King of the North’ relative to ‘Valiant’, it emerged in the spring largely unscathed based on trunk survival observations.”

  • 238: Check grammar "was not only been damaging"

Removed: been; “…..was not only damaging to grapevine dormant….”

  • 247: check grammar and context. what is "surviving" mean. Its no clear to me that any plants fully died. 

Edited surviving to trunk survival “…two individuals with high rates of trunk survival…”

  • 270: I'm not sure the big emphasis on labrusca here, as these are clearly riparia too!

Response: All focus is on riparia derived genotypes in our region; our own germplasm efforts make use of riparia heavily. This was a peculiar test winter, while we would have expected “ultra-hardy riparia-esque” genotypes like ‘Frontenac’, DM 8521, LxT 10, and the numerous riparia background-MN lines to perform above and beyond the bulk- we ultimately observed the only surviving vines to be from both V. riparia and V. labrusca heritage.

Adjustment: “…. in this study, the highest survival was observed for hybrid grapevines containing both V. riparia and V. labrusca derived characteristics.”

  • 278: produced -> developed, bred

Adjustment: produced changed to “bred”

  • 282: does it need to be in must? does it need to be at harvest? these seem like just "berry' attributes

Adjusted to read: These are both juice and jam type, noir grapevines from Elmer Swenson’s breeding efforts with characteristic V. labrusca aromas.

  • 287: "surpassing defintions" who's? what does this mean?

Removed statement to make reference cleaner, now reads: Thick wood has been shown to be less cold-hardy when examining ‘Cabernet Sauvignon’; bull-canes had larger surface area of pith, xylem, and phloem, larger xylem and phloem vascular transport units (VTUs), and more VTUs [28].

  • 295: bull wood again

Adjusted: wood replaced with cane

  • 303: "recently developed growth-regulators" is that a technique? a product used on grapes? I think this sentence needs clarification

Adjusted to read: Techniques to inhibit vigor of grapevines, such as under-vine cover cropping, via reducing available water and nutrients may also aid in slowing overly vigorous genotypes [31].

  • 314: simply "yield loss"

Adjusted as suggested.

  • 317: "reduced-to-negligible" just "reduced"

Adjusted as suggested.

  • 323: "a delay before yield returned to acceptable levels"; this is awkward. How long of a delay? hours? years? Comparing Vinifera here seems out of touch with the types of grapes in this study. any evidence from hybrids in the literature?

Adjusted: “… a multi-year delay before yield returned to commercially acceptable levels.”

Response: When searching the literature on the topic of vine renewal and retraining, hybrids have been largely untouched by researchers in applied studies. This may due to hybrids’ strong performance in all but the most challenging climates. Thus, we were forced to rely on work by researchers focusing on Vinifera in challenging environments (ex. Ohio, Washington) as examples to gain some insights.

  • 326: "ever on-going" could just be "on-goin"

Adjusted: “ever on-going” to “constant”

  • All references need to be checked. Principally, make sure to italicize scientific names. 

Adjustment: Thank you, we caught at least 2 instances of un-italicized species names.

Reviewer 2 Report

see joined document

Author Response

Reviewer 2

 Introduction

  • Line 50-51. Another method often uses to protect grapevine from cold temperatures in Canada is the geotextile. As it is an efficient winter protection method considered by producers from Québec and Ontario, the authors must mention it as an optional method, such as covering with soil or snow.

The introduction may be improved with a paragraph on cold hardiness related to genotypes characteristics (for example: genotypes process acclimation, sugars accumulation, vigor…). What are the main characteristics of cold-hardy genotypes (productivity, disease resistance, vigor, aromas, products…)? Why did we hybridization to have these genotypes?

Adjustment: included geotextiles and straw as other methods of protection.

Added: basic discussion of dormancy/hardiness.

“North Dakota grapevine production consists of cold-hardy interspecific hybrid grapevines, typically with a focus on wine grapes [9]. The short growing season and intense winter conditions have led to regional adoption of cultivars with high mid-winter freezing tolerance, with a high rate of V. riparia in their backgrounds.”

Grapevines require acclimation and winter survival of compound buds, xylem, and phloem in order to consistently yield. Grapevine acclimation is a complex process in which green, non-hardy tissues, transition to woody, cold-hardy tissues following protein, carbohydrate, membrane, and hormonal driven changes within cells of specific tissues [4]. Cane material (xylem and phloem) and dormant buds undergo dormancy acclimation processes as the growing season approaches an end, these include bud ripening and periderm development [4,8]. In fall and early winter, tissues increase in relative cold-hardiness before proceeding through deacclimation processes as spring warming conditions arise [4]. “

Methods

  • Give the list of grape genotypes. It could also be interesting to have a table with significant characteristics for each genotype related to cold hardiness and vigor. Maybe this information could be added to the first paragraph of the method with the experimental vineyard description. Indicated which genotypes were used for bud hardiness survey and which one for trunk renewal experience.- ok, we have some information in the results section. A table is presented in the results, it is good, and it could be interesting to add a column to indicated cold hardiness temperature. However, in the methods, it can mention that you have evaluated 12 genotypes for the bud hardiness survey (line 81), and 35 genotypes for the winter injury trial (line 100). Also indicated that all genotypes are hybrids (or cold-hardy hybrids)
  • Give more detail on cultural practices in the vineyard and characteristics of the vineyard: training mode, phytosanitary treatments (conventional, integrated pest program, biological?), interrow cover crop, ceps interspace, row space, soil type…

Adjustment- included description of planting site and vine maintenance, also added text to indicate period of sampling for dormant canes.

-“In 2004, a cold-hardy hybrid grapevine variety trial was established at the NDSU HRF, near Absaraka, ND, USA. From 2004 to 2009, nearly 40 own-rooted genotypes (both released cultivars and breeding germplasm) were planted for evaluation; however, only 30 were retained through the initiation of this experiment in 2018. Between 2012 and 2017, additional own-rooted genotypes were planted for characterization in rows adjacent to the earlier established (2004-2009) variety trial planting; of these newly planted genotypes, five recently released cultivars were evaluated in this experiment. The entire unirrigated experimental vineyard plot was planted on a 0 to 2 percent slope in Warsing, sandy loam soil. Vines were trained to multiple trunks with spur-pruned, bilateral cordons on a single high-wire at 1.8 m above the soil level. Rows 83 m long were established in a North to South orientation with a spacing of 2.4 m between vines in-row and 3.0 m between rows. A 0.5 m weed-free strip was maintained below the vine rows using tillage combined with pre-emerge (Flumioxazin, Chateau ®, Valent USA, San Ramon, CA, USA) and post-emerge (Glufosinate, Rely ®, BASF, Florham Park, New Jersey, USA) herbicide applications. Vineyard mid-rows were planted with red fescue (Festuca rubra). All grapevines genotypes were planted into four-vine experimental units in a randomized complete block-design with four replicates clustered by period of planting. A total of 16 vines were examined per replicated genotype; an additional nine unreplicated genotypes (two vines per plot, one plot) were also examined for trunk survival and retraining metrics. Evaluations of grapevine cold-hardiness were conducted for 12 cultivars through the 2018-2019 dormant season, and evaluations of grapevine trunk survival and retraining were conducted for 35 individuals in the 2019 growing season.

-“Dormant canes were collected twice monthly for a subset of twelve cultivars from early Nov. 2018 through mid Feb. 2019.”

Note: In drafting the manuscript, we also wanted to “add a column to indicated cold hardiness temperature.” However, our experiences with vine performance in ND are often contradictory to those published. This, combined with the inconsistency of published minimum temperature tolerances led us to omit direct statement of cold-hardiness for individuals. However, we are actively working on a review of this data for potential META analysis of environmental conditions.

  • Line 82. Why do you stop the bud survey at the end of January as you expected severe cold-injuries and the winter was not finished? Explain/justify- ok we have the explanation in the results (line 150-155), but LTE was done until mid-February, modifying the method accordingly.

Response: edited to account for this in M&M;

“Dormant canes were collected twice monthly for a subset of twelve cultivars from early Nov. 2018 through mid Feb. 2019.”

  • Line 99, 104. Winter injuries on the trunk were observed on how many grapevines (ceps) (it’s indicated ‘the total’, but what we have as ‘total’? give a scale ex:10-20 ceps? More?) And on how many grape varieties? Maybe add this information to the table with genotype characteristics.

Response: Added text in winter injury results-

“Approximately 25 vines out of a total of 578 experimental vines (across replicated and unreplicated plots) exhibited sap flow and microbial growth on trunk wounds prior to collapse; however, no consistent trend was observed for genotype or replicate location in field (data not shown).

Note: We unfortunately started trunk removal towards renewal efforts before we thought to record trunk splitting numerically, so we missed out on the opportunity to collect this data- which may have been very useful. Thus, we were left with the three classes of trunk survival rather than detailed parsing into death/collapse descriptions. Next time we experience a catastrophic crop failure, we will prepare to conduct these observations more thoroughly.

Results

No comments

Discussion

  • Line 259. Do you have some examples (observed in the field or according to your results) to support the claim that we can observe genotypic responses?

Response: Edited to clarify that more work is needed to understand genotypic responses (if present) across conductive tissues of different grapevine individuals.

“….as supplementary metrics to decipher what variances exist in genotypic response to extreme cold-weather.”

  • Line 332. Good conclusion.

Reviewer 3 Report

Dear editor, the manuscript of Svyantek and co-authors took advantages of the cold temperature recorded during winter 2018-2019 to perform a study on the cold-hardiness of several genotypes obtaining useful information on the trunk damages, bud survival. A list of five genotypes that got fewer damages is provided with the aim to increase their use in the viticulture in North Dakota.

The manuscript is well written and I did not find any issue. Therefore, I suggest accepting the manuscript in the current form.

Author Response

The manuscript is well written and I did not find any issue. Therefore, I suggest accepting the manuscript in the current form.